# Comparison of Oncologic Outcomes between Two Alternative Sequences with Abiraterone Acetate and Enzalutamide in Patients with Metastatic Castration-Resistant Prostate Cancer: A Systematic Review and Meta-Analysis

**DOI:** 10.3390/cancers12010008

**Published:** 2019-12-18

**Authors:** Doo Yong Chung, Dong Hyuk Kang, Jong Won Kim, Do Kyung Kim, Joo Yong Lee, Chang Hee Hong, Kang Su Cho

**Affiliations:** 1Department of Urology, Inha University School of Medicine, Incheon 22212, Korea; wjdendyd@gmail.com (D.Y.C.); dhkang0424@inhauh.com (D.H.K.); 2Department of Urology, Urological Science Institute, Yonsei University College of Medicine, Seoul 03722, Korea; 3Department of Urology, Gangnam Severance Hospital, Yonsei University College of Medicine, Seoul, 211 Eonju-ro, Gangnam-gu, Seoul 06273, Korea; doctor2play@yuhs.ac (J.W.K.); chhong52@yuhs.ac (C.H.H.); 4Department of Urology, Soonchunhyang University Medical College, Soonchunhyang University Seoul Hospital, Seoul 04401, Korea; dokyung80@hotmail.com; 5Department of Urology, Severance Hospital, Yonsei University College of Medicine, Seoul 03722, Korea; joouro@yuhs.ac

**Keywords:** metastatic castration-resistant prostate cancer, abiraterone acetate, enzalutamide, sequential therapy, systemic review, meta-analysis

## Abstract

Sequential treatment of androgen receptor axis targeted agents (ARAT), abiraterone acetate (ABI) and enzalutamide (ENZA), in metastatic castration-resistant prostate cancer (mCRPC) demonstrated some positive effects, but cross-resistances between ABI and ENZA that reduce activity have been suggested. Therefore, we conducted a meta-analysis to compare oncologic outcomes between the treatment sequences of ABI-ENZA and ENZA-ABI in patients with mCRPC. The primary endpoint was a combined progression-free survival (PFS), and the secondary endpoint was overall survival (OS). A total of five trials on 553 patients were included in this study. Each of the included studies was retrospective. In two studies including both chemo-naïve and post-chemotherapy mCRPC patients, for ABI-ENZA compared with ENZA-ABI, pooled hazard ratios (HRs) for PFS and OS were 0.37 (*p* < 0.0001; 95% confidence intervals (CIs), 0.23–0.60) and 0.64 (*p* = 0.10; 95% CIs, 0.37–1.10), respectively. In three studies with chemo-naïve mCRPC patients only, for ABI-ENZA compared with ENZA-ABI, pooled HRs for PFS and OS were 0.57 (*p* = 0.02; 95% CIs, 0.35–0.92) and 0.86 (*p* = 0.39; 95% CIs, 0.61–1.21), respectively. The current meta-analysis revealed that ABI-ENZA had a significantly more favorable oncological outcome, but the level of evidence was low. Therefore, large-scale randomized trials may be needed.

## 1. Introduction

Prostate cancer (PC) is the most common type of newly diagnosed malignancy in men, accounting for nearly 30% of all diagnosed cancers in men [1]. The initial treatment of patients with advanced PC typically includes hormonal therapy along with medical or surgical castration (with or without antiandrogens) [2,3]. However, most metastatic PCs acquire resistance to initial hormonal therapy in approximately 2–3 years and progress to castration-resistant prostate cancer (CRPC) [4]. Historically, PC was considered moderately resistant to cytotoxic therapy [5]. A decade ago, docetaxel was the only agent that improved the overall survival (OS) of metastatic CRPC (mCRPC) [6].

It has been well-documented that the androgen receptor (AR) axis is a key factor that drives the disease progression of PC, even after the acquisition of a castration-resistant phenotype [7]. Consequently, two agents were developed as novel AR-axis-targeted (ARAT) agents—the androgen biosynthesis inhibitor abiraterone acetate (ABI) and the second-generation antiandrogen enzalutamide (ENZA)—and they improved survival outcomes in pivotal randomized clinical trials (RCTs) for both docetaxel-naïve and treated patients with mCRPC [8,9,10,11]. In addition, these novel ARAT therapies have better tolerability with respect to hematologic problems than docetaxel chemotherapy [12].

Although these two ARAT agents have similar clinical efficacy, no study has directly compared the clinical outcomes of ABI and ENZA, and there exists no biomarker study aiding the individualized selection of ARAT agents. In contrast, several studies have evaluated the outcomes of the second ARAT agent after treatment failure of the first ARAT agent because their mechanism of action is different [13]. Sequential treatment of these ARAT agents demonstrated some positive effects, but cross-resistances between ABI and ENZA that reduce activity has been suggested [14,15]. Recently, some studies comparing the sequence of ABI to ENZA or ENZA to ABI have been published [16,17,18,19,20]. However, they were retrospective series with a small number of patients and inconsistent results. Thus, a pooled analysis of these studies was necessary. Therefore, we compared the clinical outcomes between the treatment sequences of ABI–ENZA and ENZA–ABI through a systematic review and meta-analysis for determining the appropriate treatment sequence of these therapies in patients with CRPC.

## 2. Materials and Methods

This systematic review was registered in PROSPERO (CRD42019142534).

### 2.1. Search Strategy

A literature search of all publications up to July 2019 was conducted using the Embase, PubMed, and Cochrane library databases. In addition, a cross-reference search of eligible articles was performed to detect studies that were not found in the computerized search. We used combinations of the following MeSH terms and keywords such as “prostate,” “cancer,” “carcinoma,” “abiraterone,” “enzalutamide,” “sequencing,” “sequential,” and relevant variants. The search included articles. Two authors (DYC and JWK) independently reviewed the titles and abstracts based on the inclusion criteria and reviewed the identified articles. In case of disagreement regarding the inclusion of an article, it was discussed with the third author (KSC).

### 2.2. Inclusion Criteria and Study Eligibility

The eligibility of a study was evaluated based on the participants, intervention, comparator, outcome, and study design approach and the Preferred Reporting Items for Systematic Reviews and Meta-Analyses (PRISMA) guidelines. We defined participants and intervention as patients with CRPC and those who underwent ABI–ENZA schedule from the beginning without any changes to the schedule, respectively. The comparator was defined as patients who initially underwent ENZA–ABI schedule. The primary endpoint was a combined progression-free survival (PFS). The secondary endpoint was overall survival (OS). Combined PFS is defined as: (1) the time from the start date of the first-line treatment with the first ARAT agent to disease progression following the second ARAT agent; or (2) the sum of PFS1 and PFS2, representing PFS on the first- and second-line ARAT agents. Disease progression was defined as prostate-specific antigen (PSA) and/or radiographic progression examined using the Prostate Cancer Working Group (PCWG) 2 or 3 criteria [21,22]. OS was defined as the time from the initiation of the first ARAT agent to death from any cause. There was no restriction on research design, and both randomized controlled and nonrandomized observational studies were included. The exclusion criteria were as follows: (1) non-human study, (2) not written in English, (3) conference and meeting abstracts, and (4) unable to extract outcome data. Conference and meeting abstracts were excluded even if they fit the inclusion criteria for reducing publication bias.

### 2.3. Data Extraction

Two authors (DYC and JWK) independently reviewed the included articles and extracted data at the trial level for each trial. Any discrepancy in extracted data was resolved through consensus. Extracted data included details on study design, inclusion and exclusion criteria, whether participants were randomized or nonrandomized, participant demographics and oncological characteristics, patient treatment characteristics (ABI–ENZA or ENZA–ABI in patients with CRPC), outcomes measured (combined PFS and OS), hazard ratios (HRs), 95% confidence intervals (CIs), and *p*-values.

### 2.4. Study Quality Assessments

After the final group of articles was agreed upon, two authors (DYC and JWK) independently examined the quality. Quality evaluation of the nonrandomized studies was performed according to the Newcastle–Ottawa Scale (NOS) [23]. The three major assessment categories of the NOS were selection, comparability, and exposure. A study can be given a rating of up to 9 stars, and a final score of 6 stars or more indicated a high quality. In addition, we assessed the quality of the generated evidence using the Grading of Recommendations, Assessments, Developments, and Evaluation (GRADE) system. GRADE is used to systematically approach the evaluation and strength of recommendations. It consists of domains for methodology evaluation, accuracy of results, consistency of results, immediacy, and risk of publication bias. Based on these five criteria, the quality of evidence was rated as belonging to one of four levels (high, moderate, low, and very low) [24].

### 2.5. Statistical Analysis

The effects of ABI–ENZA compared with ENZA–ABI were measured using HRs. Log HR values were obtained directly from the trials reporting HR point estimates and CIs, and the standard errors of log HR were calculated using the published CIs [25]. In addition, PFS and OS analyses were performed for both chemo-naïve patients alone and chemo-naïve and post-chemotherapy patients. Estimates for the included studies were then combined using a random-effects model with inverse variance. Pooled HRs with 95% CIs indicated the effects of ABI–ENZA or ENZA–ABI on OS and PFS. Chi-square heterogeneity tests were used to test for statistical heterogeneity between trials. The I^2^ statistic was calculated to measure discrepancies between clinical trials. A Cochran Q statistic *p*-value < 0.05 or an I^2^ statistic > 50% was used to indicate the presence of statistically significant heterogeneity between clinical trials [26]. Because fewer than 10 studies qualified for this review, funnel plots were not used to assess small study effects. Sensitivity analysis was performed by evaluating the stability of results by sequentially excluding each included study. We used Review Manager v.5.3 (2008; Nordic Cochrane Center, Cochrane Collaboration, Copenhagen, Denmark) for performing the meta-analysis. All *p*-values were two-sided, and except for the test of discrepancy, a *p*-value <0.05 was considered statistically significant.

## 3. Results

### 3.1. Systematic Review Process

The results for the PRISMA flow diagram are presented in Figure 1. The initial database search found 413 studies (137 in Embase, 238 in PubMed, and 38 in the Cochrane library). Among them, 53 studies remained after duplicates were removed. After reviewing the titles and abstracts, 23 articles were excluded. Subsequently, analysis of the full-text articles was performed based on pre-established inclusion criteria. Finally, five studies with a total of 553 patients were included. Information on the included studies is presented in Table 1. All five studies were retrospective case–control studies [16,17,18,19,20]. Three were conducted in Japan [17,18,20], one in the United States [19], and one in Japan and the United States [16]. Patients enrolled in the Maughan et al. study were also participants in the Terada et al. study [16,19]. All trials enrolled patients diagnosed with CRPC who had undergone ARAT treatment according to either ABI–ENZA or ENZA–ABI. In addition, the studies of Maughan et al. and Mori et al. included both chemo-naïve and post-chemotherapy CRPC patients [17,19]. The other studies were performed only in chemo-naïve CRPC patients [16,18,20].

### 3.2. Quality Assessment

The results of quality assessment using NOS for the included studies are shown in Table 2. All studies received a score of 6 points. Overall, quality scores within subscales represented a relatively high quality. However, in all studies, there was a bias for the selection of the control group cases.

### 3.3. Progression-Free Survival

A. Both chemo-naïve and post-chemotherapy CRPC patients:

In two studies that included both chemo-naïve and post-chemotherapy CRPC patients, meta-analysis for PFS revealed an overall adjusted HR of 0.37 for ABI–ENZA compared with ENZA–ABI (*p* < 0.0001; 95% CIs, 0.23–0.60). No heterogeneity was identified across studies (Cochran Q statistic, *p* = 0.93; I^2^ statistic, 0%) (Figure 2a).

B. Chemo-naïve CRPC patients only:

In three studies with chemo-naïve CRPC patients only, meta-analysis for PFS demonstrated an overall adjusted HR of 0.57 for ABI–ENZA compared with ENZA–ABI (*p* = 0.02; 95% CIs, 0.35–0.92). There was a statistically significant heterogeneity across studies (Cochran Q statistic, *p* = 0.02; I^2^ statistic, 74%) (Figure 2b).

### 3.4. Overall Survival

#### 3.4.1. Both Chemo-Naïve and Post-Chemotherapy CRPC Patients:

In two studies that included both chemo-naïve and post-chemotherapy CRPC patients, there was no significant difference in OS between ABI–ENZA and ENZA–ABI groups (adjusted HR, 0.64; 95% CIs, 0.37–1.10; *p* = 0.10). No heterogeneity was identified across studies (Cochran Q statistic, *p* = 0.58; I^2^ statistic, 0%) (Figure 3a).

#### 3.4.2. Chemo-Naïve CRPC Patients:

In three studies with chemo-naïve CRPC patients only, there was no significant difference in OS between ABI–ENZA and ENZA–ABI groups (adjusted HR, 0.86; 95% CIs, 0.61–1.21; *p* = 0.39). No heterogeneity was identified across studies (Cochran Q statistic, *p* = 0.89; I^2^ statistic, 0%) (Figure 3b).

### 3.5. The Quality of Evidence using the GRADE

The assessment of the quality of evidence of each comparison using the GRADE approach is shown in Table 3. Certainty was “low” in 1 comparison, and “very low” in the others.

## 4. Discussion

Since 2010, five novel agents—including ABI, ENZA, sipuleucel-T, radium-223, and cabazitaxel—have been approved for patients with CRPC. Although the survival benefits of these agents have been established through phase 3 trials, no evidence exists regarding their clinical activity when these drugs are given sequentially [15]. Thus, studies on the optimization of the treatment sequence of these agents are necessary. During the last two decades, docetaxel chemotherapy has been used in patients with mCRPC [6], but its use may be limited due to the modest survival benefit and considerable side effects as a cytotoxic agent [27,28]. ABI and ENZA are ARAT agents with advantages of oral administration, fewer adverse events than docetaxel chemotherapy, and improved oncologic outcomes in patients with CRPC in both pre- and post-chemotherapy settings [8,9,10,11]. Therefore, ARAT agents are currently chosen as the first agent in most patients with CRPC. In addition, a second ARAT agent can be used subsequently instead of chemotherapy in case disease progression occurs after first drug use. Several retrospective studies have analyzed the outcomes of sequential treatment with ABI and ENZA in either order [29,30,31,32,33,34,35,36]. Several studies demonstrated a very low PSA response rate (around 10%) [30,31,32,33], which suggests a cross-resistance between the two agents, thus raising some doubt about the clinical benefit of a sequential therapy regimen with ABI and ENZA [37]. However, other studies reported an acceptable PSA response rate (around 40%) [35,36], which suggests that sequential treatment of ARAT agents can be beneficial in a certain group of patients. The optimal sequence of these drugs has been questioned, and several comparative studies on this topic have been published with inconsistent results [16,17,18,19,20].

The present study demonstrated better PFS outcomes of ABI–ENZA than ENZA–ABI, regardless of the previous chemotherapy status but no difference in OS outcomes between the two groups, suggesting that ABI–ENZA sequence can be helpful in maximizing the likelihood of achieving acceptable PSA responses with both agents. Further, the results imply that this sequence exhibited reduced cross-resistance, which can be a plausible explanation for the difference in PFS outcomes [19]. Two previous studies have compared the sequential treatment of ARAT agents [13,38]. Maines et al. compared the cumulative monthly survival rates between ABI–ENZA (7 studies, 317 patients) and ENZA–ABI (3 studies, 80 patients) [13] and suggested that ENZA–ABI led to a slightly longer OS than ABI–ENZA. In addition, Zhang et al. analyzed differences in oncological outcomes between these two sequences through indirect comparisons [38]. The median OS was 9.7 and 7.4 months and median PFS was 3.2 and 2.9 months in ENZA–ABI and ABI–ENZA, respectively. The results of these two studies are contrary to our findings, but there are significant differences between our study and their studies. First, our study predominantly comprised docetaxel-naïve CRPC patients (90%), whereas their studies included patients with a history of docetaxel chemotherapy failure. Second, our study was a meta-analysis that incorporated direct comparisons, whereas their results were drawn from indirect comparisons. In addition, we calculated the estimate of pooled effect with adjusted HRs from multivariate analysis, whereas they used unadjusted HRs from univariate analysis.

Chopra et al. performed indirect comparisons between ABI and ENZA in both pre-docetaxel and post-docetaxel settings based on published phase 3 RCTs [39] and found that ENZA outperformed ABI with prednisone in terms of radiographic PFS, time until PSA progression, and PSA response rate but not OS in both the pre- and post-docetaxel settings. The rate of adverse events of grade ≤3 was similar between ABI and ENZA. Therefore, it might be more reasonable to choose ENZA than ABI when ARAT agents are not used sequentially. However, for the sequential use of ARAT agents, the ABI–ENZA sequence might be a better choice than the ENZA–ABI sequence, especially in docetaxel-naïve CRPC patients, based on our findings. The lack of randomized head-to-head comparison data between ABI and ENZA and between ABI–ENZA and ENZA–ABI makes it difficult to choose the optimal first-line treatment either pre- or post-chemotherapy in patients with CRPC. An ongoing, prospective RCT comparing the use of ABI–ENZA and ENZA–ABI (clinical trial information: NCT02125357) will provide valuable information for optimal sequencing of ARAT agents. Apart from the oncologic outcome, the use of ABI–ENZA may have the advantage of a lower cost of treatment than ENZA–ABI in patients with mCRPC [40]. The duration of the first ARAT agent is generally longer than that of the second drug due to cross-resistance; therefore, the ABI–ENZA sequence may reduce the burden of pharmacy cost on patients compared to ENZA–ABI. However, this may vary depending on the health insurance system of each country.

Another critical issue is whether to choose docetaxel chemotherapy or ARAT therapy first. Recently, biomarker studies have been conducted to predict treatment response and prognosis. Among these, androgen receptor-variant 7 (AR-V7) appears to be a promising molecular biomarker that predicts the response to ARAT agents and taxane treatment [41,42]. AR-V7-positive patients showed a lower response rate with ENZA or ABI than AR-V7-negative patients. Although the presence of AR-V7 is not an absolute contraindication for ARAT agents, it may reduce the effectiveness of these drugs. Taxane therapy is more efficacious than ARAT agents in AR-V7-positive patients. However, the clinical outcomes did not seem to differ significantly on the basis of the type of therapy used in AR-V7-negative patients [43]. In addition, several studies suggested that docetaxel chemotherapy may produce resistance to AR [14,15]. Therefore, ARAT therapy may be an appropriate option in chemo-naïve patients for adequate treatment responses, especially when the AR mutation status is not identified.

There are some limitations to our study. First, all five studies included in our analysis were retrospective studies, there was unavoidable bias such as nonrandomized treatment allocation, unbalanced grouping, and non-standardized data collection. This is a major drawback of our study. So, we provided the quality of evidence for the synthesized outcome through the GRADE approach. As a result, the level of evidence was low or very low, owing to the nature of retrospectively designed studies and small number patients. Thus careful interpretation is required, and large-scale randomized studies are needed to verify this result. Second, studies by Maughan et al. and Mori et al. included both chemo-naïve and post-chemotherapy patients. As mentioned above, the efficacy of ARAT drugs can be limited in the post-chemotherapy setting. Third, there were some differences in the definition of combined PFS and disease progression between the studies. Two studies evaluated disease progression only with PSA, and some patients may have radiographic progression without PSA progression. Finally, data on subsequent treatment after sequential therapy of ARAT agents were not collected; therefore, their impact on the outcomes was not considered. Despite these limitations, our study is valuable as the first meta-analysis that directly compares the oncologic outcomes between ABI–ENZA and ENZA–ABI. We provide evidence that ABI–ENZA may be a better sequential ARAT treatment than ENZA–ABI in patients with mCRPC. This finding will be useful until more solid conclusions through well-designed RCTs can be obtained.

## 5. Conclusions

In the present meta-analysis, a significant difference was identified in the oncological outcomes between ABI–ENZA and ENZA–ABI in patients with mCRPC. ABI–ENZA had better PFS outcomes than ENZA–ABI; however, there was no difference in OS outcomes. ABI–ENZA might be an optimal strategy for sequential ARAT treatment in patients with mCRPC based on the currently available evidence, but this level of evidence was low or very low. Therefore, the interpretation of this result should be undertaken carefully, and large-scale randomized trials are needed to verify this result.

## Figures and Tables

**Figure 1 cancers-12-00008-f001:**
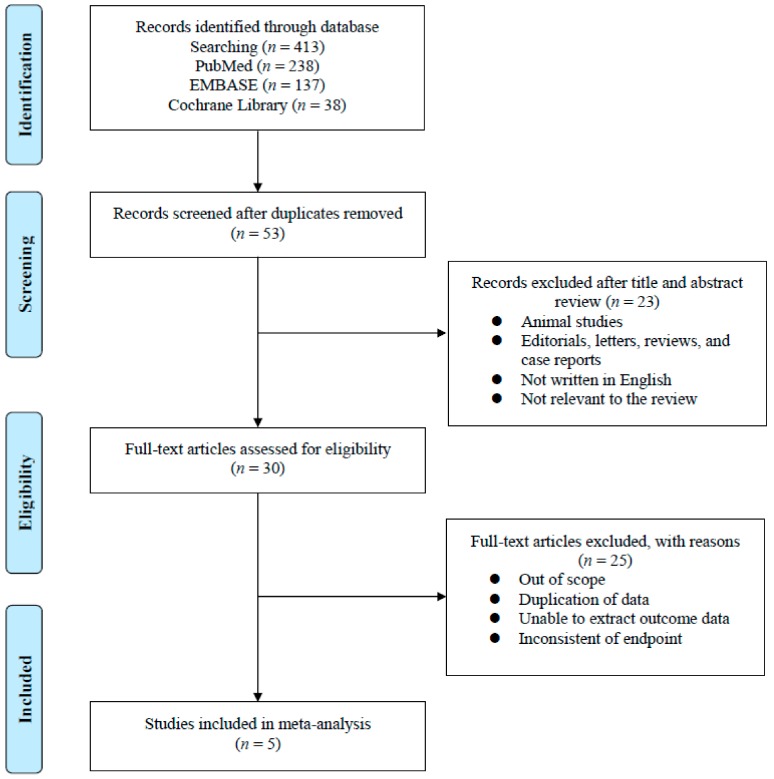
Flowchart for the systematic review process and data acquisition.

**Figure 2 cancers-12-00008-f002:**
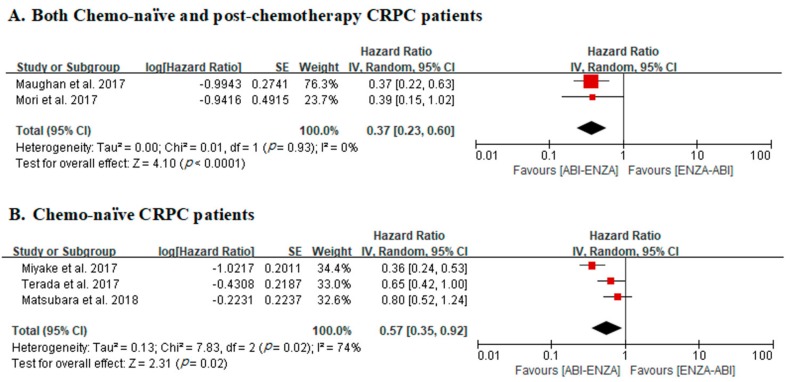
Forest plots for progression-free survival in studies including both chemo-naïve and post-chemotherapy castration-resistant prostate cancer (CRPC) patients (**A**) and chemo-naïve CRPC patients only (**B**).

**Figure 3 cancers-12-00008-f003:**
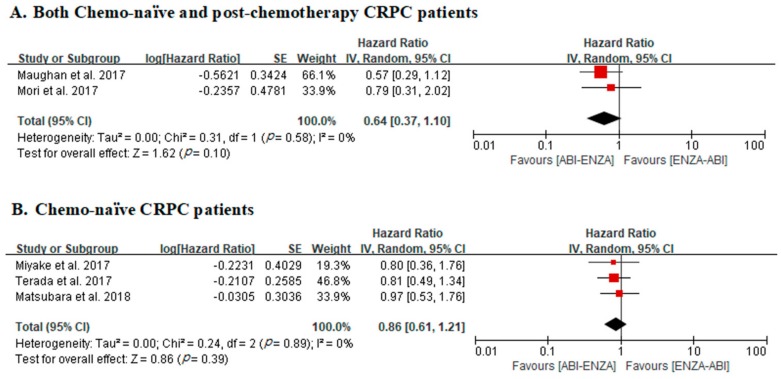
Forest plots for overall survival in studies including both chemo-naïve and post-chemotherapy CRPC patients (**A**) and chemo-naïve CRPC patients only (**B**).

**Table 1 cancers-12-00008-t001:** Characteristics of the eligible studies.

Author(s) (Year)	Country	Study Design	Study Period	Study Summary	1. Definition of Combined PFS 2. Definition of Disease Progression 3. Definition of OS	Schedule	No. of Patients	PFS (Months, IQR)	OS (Months, IQR)
Maughan et al. (2017)	USA	Retrospective	Since 2011	Comparisons of oncologic outcomes between ABI-to-ENZA and ENZA-to-ABI in both chemo-naïve (*n* = 58) and post-chemotherapy (*n* = 23) mCRPC patients	1. Sum of PFS1 and PFS2, representing PFSs on the first- and second-line ARAT agents 2. PSA and/or radiographic progression examined using the PCWG2 criteria 3. The time from initiation of the first ARAT agent to death from any cause	ABI-to-ENZA	65	19.53 (15.5–22.3)	33.27 (25.4–NR)
						ENZA-to-ABI	16	13.02 (10.3–21.2)	29.98 (18.8–NR)
Mori et al. (2017)	Japan	Retrospective	2014–2016	Comparisons of oncologic outcomes between ABI-to-ENZA and ENZA-to-ABI in both chemo-naïve (*n* = 37) and post-chemotherapy (*n* = 32) CRPC patients	1. Sum of PFS1 and PFS2, representing PFSs on the first- and second-line ARAT agents 2. PSA and/or radiographic progression examined using the PCWG2 criteria 3. The time from initiation of the first ARAT agent to death from any cause	ABI-to-ENZA	23	9 (NR)	24 (NR)
						ENZA-to-ABI	46	7 (NR)	24 (NR)
Miyake et al. (2017)	Japan	Retrospective	2014–2016	Comparisons of oncologic outcomes between ABI-to-ENZA and ENZA-to-ABI in chemo-naïve mCRPC patients (*n* = 108)	1. Sum of PFS1 and PFS2, representing PFSs on the first- and second-line ARAT agents. 2. PSA progression examined using the PCWG2 criteria 3. The time from initiation of the first ARAT agent to death from any cause	ABI-to-ENZA	49	18.4 (NR)	NR
						ENZA-to-ABI	59	12.8 (NR)	22.1
Terada et al. (2017)	USA and Japan	Retrospective	Since 2011 in USASince 2014 in Japan	Comparisons of oncologic outcomes between ABI-to-ENZA and ENZA-to-ABI in chemo-naïve mCRPC patients (*n* = 198)	1. From initiation of the ARAT agent until the time of progression on the subsequent ARAT agent 2. PSA progression examined using the PCWG2 criteria 3. The time from initiation of the first ARAT agent to death from any cause	ABI-to-ENZA	113	455 days (385–495)	919 days (761–NR)
						ENZA-to-ABI	85	296 days (235–358)	899 days (743–NR)
Matsubara et al. (2018)	Japan	Retrospective	2014–2016	Comparisons of oncologic outcomes between ABI-to-ENZA and ENZA-to-ABI in chemo-naïve mCRPC patients (*n* = 97)	1. From initiation of the ARAT agent until the time of progression on the subsequent ARAT agent 2. PSA and/or radiographic progression examined using the PCWG2 or PCWG3 criteria 3. The time from initiation of the first ARAT agent to death from any cause	ABI-to-ENZA	50	11.1 (8.00–14.05)	25.4 (19.8–31.1)
						ENZA-to-ABI	47	9.04 (6.84–11.24)	24.2 (20.2–28.2)

ABI, abiraterone acetate; ARAT, androgen receptor-axis targeted; ENZA, enzalutamide; mCRPC, metastatic castration resistance prostatic carcinoma; NA, not available; NR, not reached; OS, overall survival; PCWG, Prostate Cancer Working Group; PFS, progression-free survival.

**Table 2 cancers-12-00008-t002:** Results of the quality assessment of nonrandomized studies by the Newcastle–Ottawa Scale.

Author(s) (Year)	Selection (4)				Comparability (2)	Exposure (3)			Total Score
	Adequate Definition of Cases	Representativeness	Selection of Controls	Definition of Controls	Control for Important Factor or Additional Factor	Ascertainment of Exposure	Same Method of Ascertainment for Cases and Controls	Non-Response Rate	
Maughan et al. (2017)	1	1	0	0	2	1	1	0	6
Mori et al. (2017)	1	1	0	0	2	1	1	0	6
Miyake et al. (2017)	1	1	0	0	2	1	1	0	6
Terada et al. (2017)	1	1	0	0	2	1	1	0	6
Matsubara et al. (2018)	1	1	0	0	2	1	1	0	6

**Table 3 cancers-12-00008-t003:** Results of the GRADE quality assessment of direct evidence of each comparison.

Certainty Assessment							Number of Patients		Effect	Certainty	Importance
Number of Studies	Study Design	Risk of Bias	Inconsistency	Indirectness	Imprecision	Other Considerations	ABI-ENZA	ENZA-ABI	Relative (95% CI)		
**Progression-Free Survival of both Chemo-Naïve and Post-Chemotherapy CRPC Patients**											
2	retrospective observational studies	not serious	not serious	not serious	Serious ^a^	large effect ^b^	88	62	HR 0.37 (0.23–0.60)	Low	critical
**Overall Survival of both Chemo-Naïve and Post-Chemotherapy CRPC Patients**											
2	retrospective observational studies	not serious	not serious	not serious	Serious ^a,d^	none	88	62	HR 0.64 (0.37–1.10)	Very low	important
**Progression-Free Survival of Chemo-Naïve CRPC Patients**											
3	retrospective observational studies	not serious	Serious ^c^	not serious	not serious	none	212	191	HR 0.57 (0.35–0.92)	Very low	important
**Overall Survival of Chemo-Naïve CRPC Patients**											
3	retrospective observational studies	not serious	not serious	not serious	Serious ^d^	none	212	191	HR 0.86 (0.61–1.21)	Very low	important

ABI, abiraterone acetate; CRPC, castration resistance prostatic carcinoma; ENZA, enzalutamide; HR, hazard ratio; OS, overall survival; PFS, progression-free survival. ^a^ Total number of participants is small. ^b^ There is a large magnitude of effect. ^c^ Significant heterogeneity is observed. ^d^ The upper and lower limits of 95% CI include both meaningful benefit and harm.

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
