# Peer review of "Comparison of Oncologic Outcomes between Two Alternative Sequences with Abiraterone Acetate and Enzalutamide in Patients with Metastatic Castration-Resistant Prostate Cancer: A Systematic Review and Meta-Analysis"

_cancers, 2019, doi:10.3390/cancers12010008_

Round 1
Reviewer 1 Report
The authors now clearly describe the quality of evidence using the GRADE approach and are taking into account the limitations of their study, particularly the low level of evidence and the need for large-scale RCT, carefully.
Reviewer 2 Report
The authors answered my questions appropriate. I have no further comments.
Reviewer 3 Report
Authors replied to criticisms raised and the Ms has been improved.
This manuscript is a resubmission of an earlier submission. The following is a list of the peer review reports and author responses from that submission.
Round 1
Reviewer 1 Report
lines 59-60: tolerability is not the unique nor the most relevant driver for choosing ARAT versus docetaxel as first line in mCRPC (!!!)
lines 157-186: how many patients have been calculated for PFS/OS?
line 192: The reference is not cited appropriately
line 206: I think that this statement is rather confident…
lines 207-210: “The present study demonstrated better PFS outcomes of ABI-ENZA than ENZA-ABI, regardless of the previous chemotherapy status but no difference in OS outcomes between the two groups, suggesting that ABI-ENZA sequence can be helpful in maximizing the likelihood of achieving acceptable PSA responses with both agents”.
I do not agree with this uncorrect statement: If there are not differences in OS, sequences ABI-ENZA or ENZA-ABI are likely similar in terms of efficacy, and PFS or PSA response do not have the same importance in terms of efficacy as OS has.
lines 210-212: “Further, the results imply that this sequence exhibited reduced cross- resistance, which can be a plausible explanation for the difference in PFS outcome [18]. However, it is still unknown why ABI-ENZA had less cross-resistance than ENZA-ABI”.
The two sentences are contradictory
lines 236-240: “ Apart from the oncologic outcome, the use of ABI- ENZA may have the advantage of a lower cost of treatment than ENZA-ABI in patients with mCRPC [38]. The duration of the first ARAT agent is generally longer than that of the second drug due to cross- resistance; therefore, the ABI-ENZA sequence may reduce the burden of pharmacy cost on patients compared to ENZA-ABI”.
Costs are different among Countries according to several factors, firstly they depend on Health governing institution by each Country; therefore costs can not be a driver for choosing a sequence vs the other one
line 249: “In addition, docetaxel chemotherapy produces resistance to AR”.
This is just an hypothesis, but it has to be proven in phase 3 RCT before to become a statement.
Conclusions:
There are still no solid data to draw conclusions suggesting changes in clinical practice, derived from this meta-analysis. This data need to be confirmed by RCT.
Although I do appreciate the Authors' recognition of the study limits, I believe that these limits clearly undermine the conclusions drawn by Authors.
Reviewer 2 Report
In this clear and interesting manuscript by Doo Yong Chung, Dong Hyuk Kang and colleagues conducted a meta-analysis to compare oncologic outcomes between the treatment sequences of abiraterone acetate (ABI) and enzalutamide (ENZA), in metastatic castration-resistant prostate cancer patients. The Primary endpoint was a combined progression-free survival, the secondary endpoint overall survival. Overall 533 patients from five retrospective trials were included. Pooled Hazard ratios indicated that the treatment sequence of ABIATERONE followed by ENZALUTAMIDE had a more favorable oncological outcome in comparison with ENZA-ABI.
Overall, I found the study clear and good presented, the results are interesting and of clinical relevance. Major drawback of the paper is that only retrospective studies were included which therefore have clear the risk of bias due to nonrandomized treatments for example. Nevertheless, I think the study is a good and important work for the question of sequential therapy in metastatic castration resistant prostate cancer.
I have some smaller points to answers by the authors:
Figure 3, 4: Figures with Forrest plots.A better resolution of plots and larger and/or different font is essential to make the Figure easier to read.
Figure 1: I would suggest showing the numbers of excluded studies after the specific reasons. The Problem with the resolution is true for Figure 1 as well.Reviewer 3 Report
The authors provide a well written systematic review and meta-analysis on oncologic outcomes after two alternative sequences with abiraterone acetate (ABI) and enzalutamide (ENZA) in patients with metastatic castration- resistant prostate cancer and provide interesting information on an important topic. The authors found that ABI-ENZA had a significantly more favorable oncological outcome and, thus, might be a better sequential strategy than ENZA-ABI for patients with mCRPC.
The selection of studies is appropriate, methods and materials are well described and conclusions are well supported by the data. Potential clinical implications are also discussed adequately and limitations are considered carefully in the discussion.
Until data from RCT comparing ABI- ENZA and ENZA-ABI have not been published, this systematic review and meta-analysis could aid in clinical decision making processes.